# Protein Translocation Control in *E. coli* via Temperature-Dependent Aggregation: Application to a Conditionally Lethal Enzyme, Levansucrase

**DOI:** 10.3390/biom15081199

**Published:** 2025-08-20

**Authors:** Young Kee Chae

**Affiliations:** Department of Chemistry, Sejong University, 209 Neungdong-Ro, Gwangjin-Gu, Seoul 05006, Republic of Korea; ykchae@sejong.ac.kr

**Keywords:** protein translocation, signal peptide, reversible aggregation, levansucrase

## Abstract

Precise control of protein translocation is essential for synthetic biology and protein engineering. Here, we present a temperature-responsive system using elastin-like polypeptides (ELPs) to regulate the translocation of a conditionally lethal enzyme in *Escherichia coli*. The enzyme, levansucrase, whose activity becomes lethal in the presence of sucrose, was engineered with an N-terminal signal peptide and a C-terminal ELP tag. At 37 °C, the ELP tag induced intracellular aggregation of the fusion protein, preventing its secretion and allowing cell survival, as indicated by translucent colony formation. In contrast, at 16 °C, the ELP remained soluble, permitting levansucrase secretion into the medium. The resulting conversion of sucrose into levan by the secreted enzyme led to host cell death. These findings highlight ELP-mediated aggregation as a reversible and tunable strategy for regulating protein localization and secretion in *E. coli*, with potential applications in synthetic biology, metabolic engineering, and biocontainment systems.

## 1. Introduction

The precise targeting and movement of proteins within a cell, known as protein translocation, is a fundamental aspect of cellular function [1]. In both prokaryotic and eukaryotic systems, the ability to direct proteins to their correct subcellular locations is crucial for maintaining cellular homeostasis, facilitating intercellular communication, and executing numerous biochemical pathways [2]. Beyond these fundamental roles, the precise spatiotemporal control of protein localization is increasingly recognized as a powerful handle in advanced biotechnological applications, enabling intricate manipulation of cellular processes for desired outcomes. This includes fine-tuning metabolic fluxes, engineering novel cellular factories, and developing sophisticated biosensors. In biotechnology and synthetic biology, engineering protein localization is an essential strategy for optimizing biosynthetic pathways, enhancing protein yield, and implementing programmable control over cell fate [3]. The fidelity and flexibility of protein translocation mechanisms thus represent a powerful lever for designing responsive biological systems [4].

Signal peptides are short, amino-terminal sequences that act as molecular addresses, guiding proteins to their proper cellular compartments [5]. Typically, these sequences are composed of three domains: a positively charged n-region, a central hydrophobic h-region, and a polar c-region containing the cleavage site recognized by signal peptidases [6]. These structural elements are recognized by protein translocation machineries, such as the Sec or Tat systems in bacteria, which facilitate the transport of target proteins across or into membranes [7]. Errors in signal peptide recognition or function can lead to protein mislocalization, misfolding, or degradation, potentially causing cellular stress or dysfunction [2].

In *E. coli*, two major translocation systems are employed: the general secretory (Sec) pathway and the twin-arginine translocation (Tat) pathway. The Sec pathway is the primary route for protein translocation in *E. coli*. It relies on the SecB chaperone to keep the preproteins in an unfolded state, allowing them to pass through the SecYEG translocon in the inner membrane. Once the protein is translocated, the signal peptide is cleaved by leader peptidase (LepB), a type I signal peptidase [8]. In contrast, the Tat pathway transports folded proteins and is used for proteins that require cofactors or complex folding before translocation. The signal peptides for Tat substrates often contain a distinctive twin-arginine motif (RR) in the positively charged region, which is recognized by the Tat machinery [9]. The appropriate use of each pathway depends on the folding state and function of the substrate protein. Understanding the mechanisms of signal peptides in *E. coli* has significant implications for biotechnology, including the production of recombinant proteins and the development of novel antibiotics targeting bacterial secretion systems [10].

Elastin-like polypeptides (ELPs) are synthetic biopolymers inspired by the natural protein elastin, which is known for its elasticity and resilience in tissues [11]. ELPs are notable for their ability to undergo reversible phase transitions in response to temperature changes. ELPs are typically composed of repeats of the pentapeptide motif, Val-Pro-Gly-Xaa-Gly (VPGXG), where Xaa represents any amino acid except proline [12]. This modular design allows for precise tuning of the ELP’s phase transition temperature (T_t_), which determines the threshold at which the polymer shifts from a soluble to an aggregated state [13]. Below the T_t_, ELPs remain fully soluble in aqueous solution. Above the T_t_, they phase separate into dense, coacervate-like aggregates. These transitions are fully reversible, enabling dynamic control over the solubility and localization of ELP-fused proteins. Unlike other stimuli-responsive systems, the temperature-dependent nature of ELPs offers a facile and non-invasive method for inducing reversible changes in protein solubility, making them particularly attractive for in vivo applications where chemical induction might be undesirable or impractical.

The temperature responsiveness of ELPs has been harnessed in various applications, including controlled drug delivery, tissue scaffolding, and protein purification. Their genetically encodable nature, biocompatibility, and precise tunability make them attractive tools for synthetic biology [14,15,16]. Previous studies have demonstrated that ELPs can be fused to heterologous proteins without compromising function, and that phase separation can be used to trigger reversible aggregation in a temperature-dependent manner [17]. However, relatively few studies have focused on how ELP-mediated aggregation might influence protein translocation within living bacterial systems.

In this study, we explore the use of an ELP fusion strategy to modulate protein translocation in *E. coli* by temperature-dependent aggregation. We engineered a fusion protein comprising levansucrase—a conditionally lethal enzyme—fused at the C-terminus to an ELP tag, with an N-terminal signal peptide facilitating secretion via the Sec pathway. Levansucrase catalyzes the polymerization of sucrose into levan, a fructan that is lethal to *E. coli* when accumulated in the periplasm. Thus, controlling levansucrase localization presents an opportunity to regulate bacterial viability. We hypothesized that above the T_t_ of the ELP, intracellular aggregation would block translocation and prevent levansucrase secretion, allowing cells to survive in the presence of sucrose. Conversely, at temperatures below the T_t_, the ELP would remain soluble, permitting secretion of levansucrase and resulting in sucrose-mediated cytotoxicity.

By combining insights from protein secretion, thermal phase behavior of ELPs, and conditional lethality through enzyme localization, this study presents a synthetic biology tool to reversibly control protein translocation and function in bacteria. Previous work from our group demonstrated that proteins fused to the same ELP tag (I48), such as MBP-I48, can be reliably expressed and purified in *E. coli* without significant reduction in yield [18]. Our findings offer a foundation for future strategies in programmable cell behavior, biocontainment, and environment-responsive genetic circuits.

## 2. Materials and Methods

### 2.1. Plasmid Construction

The gene encoding levansucrase (*sacB*), originally derived from *Bacillus subtilis*, was amplified via polymerase chain reaction (PCR) using plasmid pVP80K as a template [19]. To facilitate cloning, restriction sites for SgfI and PmeI were incorporated into the 5′ and 3′ ends of the amplified gene, respectively. The resulting PCR product and the destination plasmid pVP65KR-MBP-I48 were both digested with SgfI and PmeI. The MBP gene fragment was excised from the vector using gel electrophoresis, and the *sacB* fragment was similarly purified. Ligation of the *sacB* insert into the linearized vector yielded the expression plasmid pVP65KR-SacB-I48, designed to produce the ELP-tagged levansucrase fusion protein. DNA sequencing confirmed successful insertion and proper reading frame alignment. This construct retains all the essential regulatory elements from the original pVP65KR vector, including promoter, ribosome binding site, and a C-terminal I48 ELP tag designed for thermoresponsive behavior [19,20].

### 2.2. Prediction of the Transition Temperature of the Fusion Protein

The structure of levansucrase was obtained from the Protein Data Bank (PDB ID: 1OYG) and refined by adding hydrogen atoms using MolProbity [21]. Accessible surface areas were calculated using GETAREA, simulating interactions with solvent via a 1.4 Å probe radius [22,23]. To predict the transition temperature of the ELP-fused protein, the surface index (SI) method was applied [24]. Each amino acid’s surface exposure and corresponding characteristic temperature were calculated and summed, then normalized by the total accessible surface area. Each amino acid’s contribution to the overall SI was computed based on its surface exposure and empirically assigned characteristic transition temperature. The SI was used to predict the change in T_t_ caused by ELP fusion using the following equation: ΔT_t,fusion_ = −56.4 + 0.75 × SI. This approach provided a reliable estimate of the fusion protein’s phase behavior, helping to interpret aggregation behavior observed at various temperatures.

Cell viability observation: Chemically competent *E. coli* XL10-Gold cells (genotype: Tet^r^D(mcrA)183 D(mcrCB-hsdSMR-mrr)173 endA1 supE44 thi-1 recA1 gyrA96 relA1 lac Hte [F’ proAB lacI^q^ZDM15 Tn10 (Tet^r^) Amy Cam^r^]) were prepared using standard calcium chloride-based protocols. Cells were transformed with 100 pg of the pVP65KR-SacB-I48 plasmid through heat shock: 30 min on ice, 30 s at 42 °C, and recovery on ice for 2 min. Transformed cells were incubated in 500 μL of SOC medium for 1 h at 37 °C with shaking at 200 rpm, then plated on LB agar containing 50 μg/mL kanamycin for selection. To evaluate sucrose sensitivity and temperature effects, transformed cells were also plated on LB agar supplemented with 5% (*w*/*v*) sucrose and 50 μg/mL kanamycin. Plates were incubated at either 37 °C for 24 h or at 16 °C for 5 days. Colony formation, size, and morphology were observed for each condition to assess the effects of levansucrase expression, ELP aggregation, and secretion under varying environmental conditions.

### 2.3. Crude Levansucrase Preparation and Activity Test

A single colony of XL10-Gold harboring the plasmid pVP65KR-SacB-I48 was selected from an LB agar plate and used to inoculate 1 mL of LB medium supplemented with 50 μg/mL kanamycin. This starter culture was grown overnight at 37 °C with shaking at 200 rpm. The overnight culture was then used as an inoculum for 200 mL of fresh LB medium containing 50 μg/mL kanamycin, with an inoculation volume of 1:200 (*v*/*v*). The larger culture was grown at 37 °C with shaking until the optical density at 600 nm (OD_600_) reached approximately 0.8. At this point, isopropyl β-D-1-thiogalactopyranoside (IPTG) was added to induce protein expression at a final concentration of 0.5 mM. The culture was incubated for additional 4 h under the same conditions. Following the incubation period, the cells were harvested by centrifugation at 5000 rpm for 30 min at 4 °C. The resulting cell pellet was resuspended in 20 mL of 10 mM Tris-HCl buffer, pH 7.4, and subjected to sonication using an ultrasonic processor to lyse the cells and release intracellular contents. Triton X-100 was added to the lysate to the final concentration of 1%, and the total volume was adjusted to 40 mL by adding 10 mM Tris-HCl buffer pH 7.4. The lysate was centrifuged at 15,000 rpm for 30 min at 4 °C to separate the soluble fraction, which was retained for downstream analyses.

To assess levansucrase activity via turbidity, 100 μL of the soluble fraction from the cell lysate was mixed with 900 μL of McIlvaine’s buffer, pH 6.0, composed of 126 mM Na_2_HPO_4_ and 37 mM citric acid, and supplemented with 0.6 M sucrose [25]. The reaction mixture was incubated at room temperature for 12 h to allow enzymatic activity and the formation of levan. Turbidity due to levan formation was assessed by measuring OD_400_ using a spectrophotometer. As a control, the plasmid pVP65KR-MBP-I48 was used to produce MBP-I48 under identical conditions. There were 3 observations for each of the levansucrase and MBP.

To assess levansucrase activity via glucose production, 1 mL of 1 M MES pH 6.0 was added to 9 mL of the soluble fraction of cell lysate, where 0.5 g of solid sucrose was dissolved to achieve a final concentration of 5%. The mixture was then incubated at 40 °C for 2 h. Insoluble materials were removed by centrifugation at 15,000 rpm for 30 min at 4 °C. The glucose produced by levansucrase was quantified using the Glucose (GO) Assay Kit following the manufacturer’s protocol (Merck, St. Louis, MO, USA). Briefly, 2 mL of assay reagent was added to 1 mL of the soluble fraction from the incubated mixture. The reaction was allowed to proceed for 30 min at 37 °C and then terminated by adding 2 mL of 6 M H_2_SO_4_. Absorbance was measured at 540 nm. There were 4 observations for each of the levansucrase and MBP.

### 2.4. Language Editing and Translation

During manuscript preparation, the author used ChatGPT (OpenAI, GPT-4, accessed on 22 May 2025) to assist with English language editing, correction of typos and grammar, and translation of text from Korean to English. All content was carefully reviewed and verified by the author.

## 3. Results and Discussion

### 3.1. Hypothesis and Rationale

The central hypothesis of this study is that levansucrase secretion—and consequently, its lethality in sucrose-containing environments—can be controlled through reversible aggregation induced by a temperature-sensitive ELP tag. The rationale stems from the biophysical properties of ELPs, which undergo sharp phase transitions based on temperature. This makes them excellent candidates for post-translational regulation in live cells. Unlike most regulatory mechanisms that rely on inducible promoters or riboswitches, this system functions entirely at the level of protein localization.

Levansucrase is known to catalyze the polymerization of sucrose into levan, a polysaccharide that exerts toxic effects on *E. coli*, primarily due to osmotic imbalance and interference with cell wall integrity [26]. This conditional lethality provides a powerful basis for engineering a switchable system where enzyme secretion can be linked to environmental triggers. Our hypothesis was that if the secretion of levansucrase could be reversibly blocked, then cell survival in sucrose-containing environments could be externally regulated. To achieve this, we utilized an elastin-like polypeptide (ELP) tag, I48, fused to the C-terminus of levansucrase. ELPs undergo temperature-dependent phase separation, remaining soluble below their transition temperature (T_t_) and aggregating above it. We reasoned that aggregation at elevated temperatures would prevent translocation of levansucrase by sterically hindering passage through the Sec translocon. As a result, the enzyme would remain intracellular and inactive in terms of sucrose conversion. Conversely, at low temperatures, the ELP would remain soluble, allowing levansucrase to be secreted into the periplasm, where levan production is known to be toxic to *E. coli*.

This strategy assumes that ELP aggregation does not interfere with the enzymatic function of the fusion protein itself, only with its cellular localization. Given previous studies showing that ELPs do not denature fused enzymes and that ELP-mediated phase separation is fully reversible, this assumption is well-founded [17,27,28]. Moreover, intracellular levansucrase activity is expected to be limited by sucrose availability, which is typically extracellular. Thus, cell viability should correlate directly with secretion efficiency, making this system a reliable reporter of protein localization.

As shown in Figure 1, our experimental setup aims to confirm whether levansucrase remains functional and intracellular when fused ELPs turn aggregates, thereby enabling *E. coli* cells to survive and grow in a sucrose-containing medium under the specified conditions. This pattern reinforces the temperature-dependent secretion hypothesis, providing a clear functional readout based on differential growth phenotypes.

Our original research aim was not to investigate protein secretion, but rather to explore protein–ligand interactions using NMR spectroscopy. Prior studies had shown that MBP fused to the elastin-like polypeptide (I48) retained its ability to bind maltose and attenuate its NMR signal, while showing no interaction with other sugars such as sucrose, glucose, or lactose. Building on this, we intended to use levansucrase as a new protein target to observe ligand binding and NMR signal changes with sucrose. As part of this plan, we constructed the plasmid pVP65KR-SacB-I48, replacing the MBP gene in the parent vector with sacB, and retaining the C-terminal I48 tag.

However, during preliminary validation steps, we observed an unexpected result: *E. coli* harboring the pVP65KR-SacB-I48 plasmid formed colonies even on sucrose-containing plates at 37 °C—a condition under which native *sacB* expression is known to be lethal due to levan accumulation. Initially, we assumed a cloning error or expression failure and repeated the transformation and plating multiple times. Yet, the surprising phenotype persisted across all replicates.

This led us to hypothesize that the fusion of I48 to levansucrase was interfering not with the enzyme’s activity per se, but with its secretion. Since levansucrase exerts toxicity only when secreted and acts extracellularly on sucrose, any defect in translocation would neutralize its lethal effect. Considering that I48 forms temperature-dependent aggregates, we proposed that aggregation at 37 °C physically hinders secretion by preventing the fusion protein from entering the Sec translocation pathway.

### 3.2. Plasmid Design and Observations

To experimentally validate the hypothesis, we used the newly constructed plasmid pVP65KR-SacB-I48. The MBP gene in the parental vector (pVP65KR-MBP-I48) was replaced with the *sacB* gene to produce a levansucrase fusion protein with an ELP tag (I48). This construct includes a standard Sec signal peptide, which is expected to guide levansucrase through the Sec translocon for secretion into the periplasm, where it exerts its function. However, the fusion of I48 at the C-terminus introduces a temperature-dependent element. At 37 °C, the I48 domain transitions into an aggregated state, likely forming a coacervate that is incompatible with the translocation machinery.

Interestingly, the SacB-I48 fusion protein was ineffective in counter-selection assays at 37 °C, in contrast to the robust toxicity observed with native SacB. These results indicate that protein aggregation likely interferes with translocation, thereby blocking secretion and attenuating toxicity. These results validated our core design principle and prompted further investigation into temperature-controlled secretion behavior.

### 3.3. Temperature-Dependent Viability and Morphological Evidence

Transformation of *E. coli* with pVP65KR-SacB-I48 followed by plating under various conditions revealed clear temperature-dependent outcomes. At 37 °C, colonies formed on LB-sucrose plates but appeared translucent (Figure 1a), a phenotype previously associated with intracellular accumulation of levan due to failed secretion [29]. This translucency is a critical indicator. It suggests that while the cells survive, their metabolism or membrane integrity may be altered, possibly due to minor intracellular levan formation or simply the presence of aggregated protein. This morphological shift provides a visual confirmation of the altered protein localization and its impact on cellular behavior under stress. At 16 °C, no colonies appeared on sucrose plates, indicating effective secretion and enzyme-induced lethality (Figure 1b). These findings support our model: I48 aggregation at 37 °C prevents levansucrase secretion, allowing cells to survive, whereas soluble I48 at 16 °C permits secretion and cell death.

An intriguing additional observation was that translucent colonies grown at 37 °C gradually expanded over time and developed an unusual three-dimensional morphology. After approximately 10 days of incubation, some colonies reached about 3 mm in both diameter and height, forming hemispherical structures that appeared to hang downward from the surface of the agar, rather than spreading laterally across it (Figure 2). This atypical growth pattern diverges from the typical radial expansion observed in *E. coli* colonies on solid media, and may result from local accumulation of levan or increased intracellular biomass, leading to gravity-driven sagging or structural swelling at the colony base. The persistence of viable cells in these structures, along with their shape, supports the idea that intracellular levansucrase retains partial activity, though not at levels sufficient to cause acute cytotoxicity.

### 3.4. Aggregation Caused by ELP

The transition temperature of I48, an elastin-like polypeptide (ELP), was previously reported to be 22 °C [30]. To estimate the transition temperature of the ELP-fused levansucrase, we employed a computational approach that incorporated accessible surface area measurements along with the characteristic transition temperatures assigned to each amino acid residue [24]. This analysis predicted the transition temperature of levansucrase-I48 to be approximately 28 °C. This prediction aligns well with our observations regarding the reduced lethality of the system. At 37 °C, I48-fused proteins underwent aggregation, thereby preventing efficient translocation and secretion. In contrast, at 16 °C, such aggregation did not occur, allowing levansucrase to be secreted into the medium and catalyze the conversion of sucrose into levan, ultimately leading to bacterial cell death. An important characteristic of I48 aggregates is their ability to undergo reversible transitions between a fully soluble state and a phase-separated coacervate state, distinguishing them from inclusion bodies, which typically form as irreversible protein aggregates in overexpressed systems. Unlike inclusion bodies, which often require denaturing conditions for solubilization, I48 aggregates retain a dynamic and responsive nature, enabling phase separation without compromising the structural integrity or enzymatic function of the fused protein [27,28]. Thus, even when I48 promotes aggregation, it does not negatively affect the activity of levansucrase or its ability to catalyze sucrose conversion. This reversible aggregation property provides valuable insight into how ELP fusion can influence protein behavior within a cellular environment, affecting both translocation and functional outcomes. Future investigations could delve deeper into the specific biophysical nature of these aggregates within the cytoplasm and their precise interaction with components of the Sec translocon, potentially revealing new insights into how large protein complexes navigate or obstruct cellular transport machinery.

### 3.5. Production and Assay of the Functional Levansucrase

As shown in Figure 2, the bacterial colonies grown under specific conditions displayed notable morphological changes. Colonies became considerably larger and exhibited a translucent appearance, resembling the opalescent effect observed in previous studies. To further investigate the viability of cells within these colonies, several translucent colonies were selected and inoculated into fresh LB medium devoid of sucrose. Upon incubation, robust bacterial growth was observed, confirming the presence of live cells within or on the surface of the translucent colonies. To assess the enzymatic activity of levansucrase, protein expression was induced, and the lysates were subjected to functional assays. The reaction mixture containing levansucrase showed a greater increase in optical density at 400 nm compared to the mixture containing maltose-binding protein (MBP), with more than a twofold difference observed (Figure 3a). Although the mixtures did not appear visibly cloudy, the increase in turbidity suggests the formation of insoluble products. Notably, a slight increase in optical density was also detected in the MBP-containing sample, which is likely due to nonspecific protein aggregation present in the soluble fraction of the lysate. To further examine the wavelength dependence of turbidity, we also extracted the OD_600_ values from the same UV-Vis spectra and plotted them in Appendix A. The OD_600_ trend closely mirrored that of OD_400_ (Figure 3a), reinforcing the interpretation that the observed increase in optical density originates from light scattering by colloidal levan aggregates. The full spectral data for all time points are provided in Appendix A (Excel format). This comparative result supports the presence of enzymatically active levansucrase in the soluble lysate fraction, capable of converting sucrose into levan under the given experimental conditions. Despite potential aggregation effects from the ELP fusion, the observed activity confirms that the functional integrity of levansucrase is at least partially retained. These results indicate that, although the ELP-tagged levansucrase forms aggregates at 37 °C, it remains catalytically active in this aggregated state. Therefore, the survival of cells on sucrose-containing plates at 37 °C is unlikely to be due to enzyme inactivation. Rather, our findings support the hypothesis that the aggregated enzyme fails to be secreted, preventing periplasmic levan synthesis and thereby avoiding sucrose-induced toxicity. The extended lag in turbidity development is unlikely due to limited catalytic efficiency, as *B. subtilis* levansucrase has been reported to exhibit *K_m_* ≈ 30 mM and *k_cat_* ≈ 135 s^−1^ [29]. Instead, we attribute the delay to the low enzyme concentration in the crude lysate, requiring extended time for polymeric levan to accumulate to colloidal concentrations sufficient for light scattering. This accumulation-dependent threshold effect is consistent with the observed kinetics of optical density increase. Levansucrase activity was assessed using a colorimetric assay for glucose detection. In this assay, glucose released from sucrose by levansucrase was oxidized by glucose oxidase, which in turn oxidized o-dianisidine, allowing absorbance to be measured at 540 nm. SacB-I48 samples exhibited clear levansucrase activity, whereas the MBP control showed minimal to no detectable activity. Although the absorbance values at 540 nm were relatively low, the difference between SacB-I48 and control samples was consistent and reproducible, indicating specific enzymatic activity. This likely reflects the low concentration of active levansucrase in the crude lysate rather than diminished catalytic efficiency. Importantly, this result supports that the enzyme retains at least partial activity in the coacervated state, suggesting that substrate access and turnover remain feasible even under phase-separated conditions.

While these findings confirm that the ELP-tagged levansucrase is functionally active, a direct comparison of its catalytic performance to a non-tagged or commercially available enzyme would be necessary to quantify any loss of activity due to coacervation. This will be addressed in future studies to better understand how phase separation influences enzymatic function, with planned investigations focusing on catalysis in the condensed or phase-separated state.

### 3.6. Interpretation of ELP-Mediated Translocation Blocking

The key to this system lies in the reversible aggregation behavior of the ELP tag. At 37 °C, the I48 tag undergoes a phase transition, forming coacervates that likely interfere with protein engagement at the Sec translocon. Although protein translocation is a complex multistep process involving signal peptide recognition, preprotein unfolding, and passage through the translocation pore, steric interference from large aggregates can effectively halt this pathway. Our results align with this model, wherein SacB-I48 fails to be secreted at elevated temperatures due to aggregation-induced retention in the cytoplasm.

The tunability of ELP transition temperatures by sequence design and environmental conditions such as salt concentration or pH suggests that this system could be further refined to operate under alternative physiological contexts. Such regulation provides a valuable strategy for biosafety applications, where containment of engineered cells is necessary, or for metabolic channeling, where enzyme localization dictates pathway efficiency.

Moreover, it is plausible that the local physicochemical environment within the cytoplasm—including ionic strength, molecular crowding, and the presence of other chaperones or proteases—modulates the aggregation dynamics of the ELP fusion protein. These factors could either enhance or mitigate the extent to which the fusion protein aggregates and blocks translocation. It is also worth exploring whether intermediate temperatures (e.g., 25–30 °C) might result in partial secretion, leading to a graded rather than binary output in terms of cellular viability. Based on the known thermodynamic behavior of ELPs, we can speculate that partial secretion may occur near the transition temperature (~28 °C). At such intermediate temperatures, the ELP tag may be only partially aggregated or exist in a dynamic equilibrium between soluble and coacervate states. In this scenario, a small amount of levansucrase may successfully translocate to the periplasm or extracellular space, potentially resulting in low levels of sucrose conversion and a correspondingly low survival rate. This is consistent with previous reports showing tunable ELP phase behavior near the T_t_ and supports the idea that the degree of aggregation—and thus secretion—is not strictly binary but rather temperature-sensitive and gradual [13]. Such tunable behavior would be particularly useful in the design of temperature-sensitive bioswitches that operate across a spectrum of physiological conditions. Future studies using fluorescently tagged ELP-fusion constructs could provide real-time insights into subcellular localization dynamics and aggregation kinetics, offering more direct evidence for the proposed model. Although we did not quantify protein expression levels in the current study, the observed yields of ELP-tagged levansucrase were sufficient for purification and in vitro activity assays. This is consistent with previous observations using I48-fused proteins in our group, which showed robust expression and solubility.

### 3.7. Broader Implications and Future Potential

The modularity and simplicity of this system make it attractive for multiple applications in synthetic biology. Unlike transcriptional inducers or metabolite-responsive systems, temperature is a universal, easily adjustable stimulus. This opens the door to designing temperature-sensitive switches for protein localization, antimicrobial peptide release, or biosensor activation.

Future improvements might involve engineering ELP variants with different T_t_ values to fine-tune the operating range of the system. For example, modifying the amino acid composition of the ELP or adjusting buffer conditions such as salt concentration could potentially shift the T_t_ to better align with specific host organisms or application environments. Furthermore, integrating this form of post-translational control with transcriptional regulatory circuits may allow for multi-layered regulation, though the feasibility and stability of such designs remain to be explored. If successful, such combined control strategies could support the development of intracellular logic gates that respond to both internal metabolic cues and external temperature changes, potentially enhancing the precision of synthetic biological systems.

It is also conceivable that this strategy could be adapted for use in other proteins or host systems, including Gram-positive bacteria or yeast strains with robust secretion capabilities. In biomanufacturing, it might be possible to apply this mechanism to delay secretion of specific enzymes until a critical biomass threshold is reached, thereby mitigating metabolic stress during growth. In therapeutic contexts, controlled secretion of immune modulators or cytokines could, in principle, improve the timing and localization of therapeutic effects. Similarly, in environmental applications, temperature-sensitive control of bioremediation enzymes might help reduce premature degradation and increase site-specific action. While these applications are speculative at this stage, the modularity and reversibility of the system provide a promising foundation for future exploration.

## 4. Conclusions

In this study, we have demonstrated a novel strategy for the reversible, temperature-dependent regulation of protein translocation in *Escherichia coli* using elastin-like polypeptides (ELPs). By fusing a conditionally lethal enzyme, levansucrase, to an ELP tag and directing its secretion via a signal peptide, we established a system in which protein localization, and consequently cell viability, could be precisely controlled through temperature modulation. At elevated temperatures, the ELP tag aggregated and blocked secretion, allowing cells to survive in otherwise lethal conditions. At lower temperatures, the ELP remained soluble, enabling secretion and inducing cell death due to the enzymatic conversion of sucrose into toxic levan.

This work highlights the power of using genetically encoded thermoresponsive elements to dynamically control protein behavior inside living cells. Our findings provide strong evidence that ELP-induced aggregation can interfere with translocation machinery, offering a new layer of post-translational control in bacterial systems. Importantly, this approach is modular, tunable, and does not require exogenous inducers or elaborate genetic circuitry, making it highly suitable for scalable applications.

The implications of this technology are significant for synthetic biology, where conditional control of protein function is critical for building responsive genetic circuits, implementing population control, and designing safe biocontainment strategies. Beyond its immediate application in bacterial systems, this strategy may also be adapted for use in other organisms, potentially expanding its utility in industrial biotechnology, therapeutics, and biosensing platforms.

Future work will explore expanding the range of ELP transition temperatures by modifying their sequence and environmental parameters, as well as testing this system in other bacterial strains and secretion pathways. Furthermore, integrating this post-translational regulation method with transcriptional control systems could yield more sophisticated, multi-level regulatory architectures. Overall, ELP-mediated translocation control presents a promising avenue for building responsive, programmable cellular systems tailored to both research and applied biotechnology contexts.

## Figures and Tables

**Figure 1 biomolecules-15-01199-f001:**
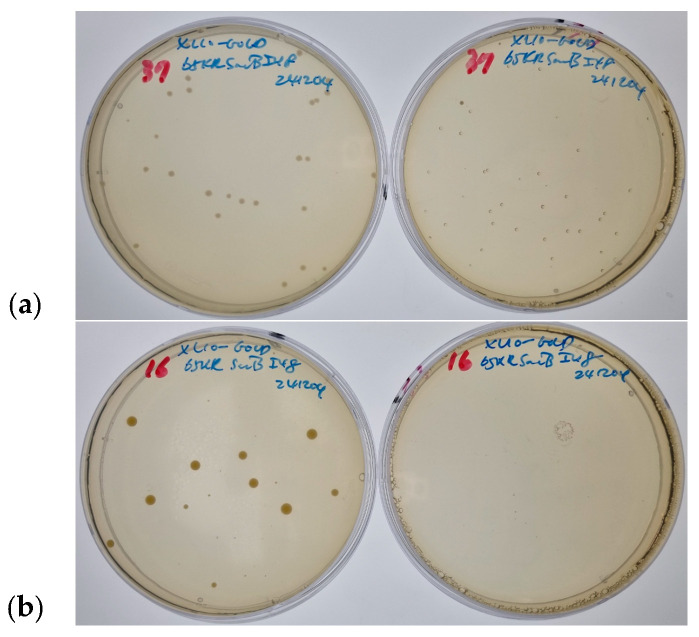
Growth of colonies on LB agar plates under different temperature and sucrose conditions. (**a**) At 37 °C (overnight incubation), colonies formed on sucrose plates appear translucent (**right**), while those on sucrose-free plates remain normal (**left**). (**b**) At 16 °C (5-day incubation), no colonies are observed on sucrose-containing plates, indicating secretion-induced lethality (**right**), while those on sucrose-free plates remain normal (**left**).

**Figure 2 biomolecules-15-01199-f002:**
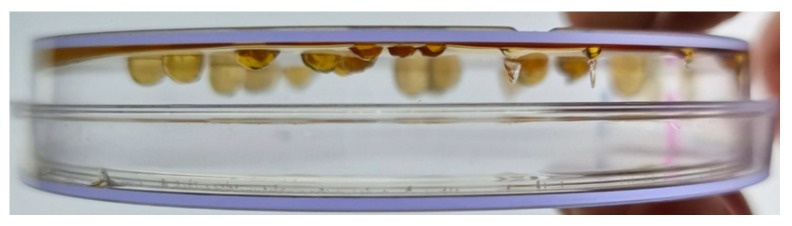
Morphology of colonies grown at 37 °C with 5% sucrose after 10 days. Colonies expanded into three-dimensional, translucent structures that hung downward from the agar surface, resembling stalactite-like formations.

**Figure 3 biomolecules-15-01199-f003:**
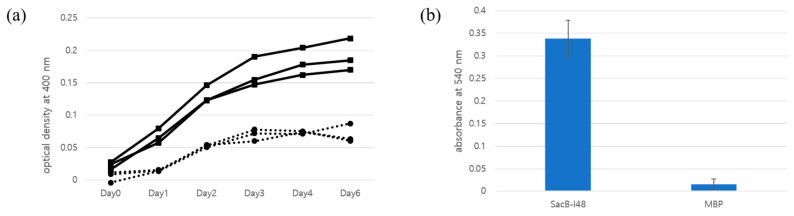
Levansucrase activity assay. (**a**) Time-dependent turbidity changes. Levansucrase samples are represented by square markers with solid lines, while MBP control samples are indicated by circle markers with dotted lines. (**b**) Colorimetric assay for glucose detection. Glucose released from sucrose by levansucrase was oxidized by glucose oxidase, accompanied by the oxidation of o-dianisidine. The resulting absorbance was measured at 540 nm. Bar heights represent the mean of four replicates, and error bars indicate the standard deviation.

## Data Availability

Data is contained within the article or Appendix A.

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
