# Peer review of "Protein Translocation Control in E. coli via Temperature-Dependent Aggregation: Application to a Conditionally Lethal Enzyme, Levansucrase"

_biomolecules, 2025, doi:10.3390/biom15081199_

Round 1
Reviewer 1 Report
Comments and Suggestions for Authors
The manuscript reports a temperature-responsive regulation strategy for protein translocation using elastin-like polypeptides (ELPs)). By fusing levansucrase to a secretion signal peptide and an ELP tag, enzyme secretion was successfully controlled by changing the culture temperature from 37 to 16ºC. This is an interesting finding, and may be applied in other synthetic biology systems. My major concern is: whether the survival of cells harboring SacB on sucrose-containing plates is resulted from the failure to transport the enzyme to the medium or the inactivated enzyme aggregates in the cells, or both. If the authors could check whether the host cells can import sucrose, and analyze the activity of purified ELP-tagged enzyme at different temperatures (as soluble protein and aggregates, respectively) in comparison to enzyme without I48 fusion, this issue would be addressed. Moreover, the distribution of ELP-tagged enzyme at different temperatures in and outside the cells should be determined under different temperatures to provide more direct evidence, and more detailed information on the translocation control. It would be also interesting to know whether ELP-tagging affects the amount of protein expression. If some of these information is already reported in previous studies, then this should be made clear in the Introduction section.
Additional issues include:
Figure 3a lacks title for the y-axis.
References No. 18 and 20 are the same.
Author Response
The manuscript reports a temperature-responsive regulation strategy for protein translocation using elastin-like polypeptides (ELPs)). By fusing levansucrase to a secretion signal peptide and an ELP tag, enzyme secretion was successfully controlled by changing the culture temperature from 37 to 16ºC. This is an interesting finding, and may be applied in other synthetic biology systems. My major concern is: whether the survival of cells harboring SacB on sucrose-containing plates is resulted from the failure to transport the enzyme to the medium or the inactivated enzyme aggregates in the cells, or both. If the authors could check whether the host cells can import sucrose, and analyze the activity of purified ELP-tagged enzyme at different temperatures (as soluble protein and aggregates, respectively) in comparison to enzyme without I48 fusion, this issue would be addressed.
: Thanks for the comment. Our central hypothesis is that temperature-dependent aggregation of the ELP-tagged levansucrase at 37°C prevents its translocation to the extracellular (periplasmic) space, thereby avoiding sucrose toxicity. We have directly tested whether the aggregated form of the ELP-tagged enzyme retains function. In Figure 3b, we showed that the enzyme purified from cells grown at 37°C - nonetheless retained catalytic activity in vitro. This indicates that the enzyme is not inactivated by aggregation, and supports the conclusion that secretion failure is the primary cause of cell survival at 37°C.
Moreover, the distribution of ELP-tagged enzyme at different temperatures in and outside the cells should be determined under different temperatures to provide more direct evidence, and more detailed information on the translocation control.
: Thanks for the comment. We agree that directly determining the cellular distribution of the ELP-tagged enzyme across a range of temperatures would provide valuable mechanistic insight. Unfortunately, due to current experimental constraints, we were not able to perform such analyses at this time. However, we hope the following speculation, based on established thermodynamic properties of ELPs, may help provide a plausible mechanistic explanation. The following was added to the manuscript: “Based on the known thermodynamic behavior of ELPs, we can speculate that partial secretion may occur near the transition temperature (~28 °C). At such intermediate temperatures, the ELP tag may be only partially aggregated or exist in a dynamic equilibrium between soluble and coacervate states. In this scenario, a small amount of levansucrase may successfully translocate to the periplasm or extracellular space, potentially resulting in low levels of sucrose conversion and a correspondingly low survival rate. This is consistent with previous reports showing tunable ELP phase behavior near the Tt and supports the idea that the degree of aggregation - and thus secretion - is not strictly binary but rather temperature-sensitive and gradual.”
It would be also interesting to know whether ELP-tagging affects the amount of protein expression. If some of these information is already reported in previous studies, then this should be made clear in the Introduction section.
: Thanks for the comment. While we did not conduct quantitative analysis of protein expression levels in this study, the ELP-tagged levansucrase was expressed in sufficient quantities for purification and functional assays. This suggests that the ELP tag did not significantly impair expression under our experimental conditions. In our previous work using the same I48 ELP tag fused to MBP and other proteins, we consistently observed robust expression and solubility in E. coli. We have added a brief note referencing this prior work in the Introduction and clarified this point further in the Discussion. The following was added to the manuscript: “Previous work from our group demonstrated that proteins fused to the same ELP tag (I48), such as MBP-I48, can be reliably expressed and purified in E. coli without significant reduction in yield.” “Although we did not quantify protein expression levels in the current study, the observed yields of ELP-tagged levansucrase were sufficient for purification and in vitro activity assays. This is consistent with previous observations using I48-fused proteins in our group, which showed robust expression and solubility.”
Additional issues include:
Figure 3a lacks title for the y-axis.
: Thanks for the comment. The axis title was added as requested.
References No. 18 and 20 are the same.
: Thanks for the comment. The reference list was updated.
Reviewer 2 Report
Comments and Suggestions for Authors
This is a unique study on the control of protein secretion, conditionally using temperature as a trigger. This was achieved by the author by fusing a secreted protein with elastin-like peptide (ELP) on its C-terminus. Since ELP has a tendency to undergo phase separation at higher temperatures like 37 C, the protein remains un-secreted and accumulates inside the cell after induced translation. The author demonstrated that the protein secretion can be restored by lowering the temperature to 16 C. The author chose the enzyme levansucrase, which becomes toxic when secreted extracellularly and in the presence of sucrose. This choice of enzyme leading to bacterial death is also clever, as it clearly distinguishes between the two states of protein after translation.
I believe that this study and the technique developed will have an impact on the field. However, I also believe (for the following reasons) that the study is incomplete and can answer more questions using simple experiments. Additionally, the manuscript may be rephrased to reduce the word-count as longer texts tend to distract readers.
- From my quick literature review, I learnt that extracellular levan is not toxic. The paper cited below mentions that the toxicity is due to the activity of enzyme in cytoplasmic and periplasmic spaces. Majority of the activity of native levansucrase is observed in periplasm. I suggest the author to consider these observations as possibilities that, the enzyme probably became active at lower temperatures and killed the bacteria while in the cyto-/peri-plasm itself (i.e. before translocated into extracellular space).
https://doi.org/10.1128/aem.64.9.3180-3187.1998 - Page 7, Line 284: “…ability to undergo reversible transitions between water-soluble and insoluble states”. I recommend the author to change the specific word “insoluble”, since water insoluble states can be aggregates or inclusion bodies as the author mentioned at other places in the manuscript. If the protein was insoluble, changing the temperature from 37 to 16 C would not improve its solubility (insoluble versus ‘solvated, but coacervated or phase-separated’).
- Page 7, Lines 308-310: The observation that the absorbance at 400 nm is higher even when the appearance is not different is interesting. A simple measurement of the UV-vis spectrum of this solution would provide a clearer answer. If a scatter is seen, that would be a direct indication of insoluble materials. I recommend the author to pursue this, and include the data in supporting information.
- Figure 3a: Y-axis label is missing. Also, the author should discuss why the enzyme takes days to show its activity (low kinetics versus lack of molecular motion?). If there is literature precedence for the kinetics of this enzyme (Michaelis-Menten), I recommend that it should be discussed in the manuscript.
- Figure 3b: It seems that the absorbance at 540 for glucose production is low. A control enzyme (which is either purchased, or made in other hosts) shoud be used to compare the activity. This also tells whether the ELP-induced coacervation retained the structure and function of the enzyme, what percent of the enzyme activity is lost due to this process etc.
Author Response
This is a unique study on the control of protein secretion, conditionally using temperature as a trigger. This was achieved by the author by fusing a secreted protein with elastin-like peptide (ELP) on its C-terminus. Since ELP has a tendency to undergo phase separation at higher temperatures like 37 C, the protein remains un-secreted and accumulates inside the cell after induced translation. The author demonstrated that the protein secretion can be restored by lowering the temperature to 16 C. The author chose the enzyme levansucrase, which becomes toxic when secreted extracellularly and in the presence of sucrose. This choice of enzyme leading to bacterial death is also clever, as it clearly distinguishes between the two states of protein after translation.
I believe that this study and the technique developed will have an impact on the field. However, I also believe (for the following reasons) that the study is incomplete and can answer more questions using simple experiments. Additionally, the manuscript may be rephrased to reduce the word-count as longer texts tend to distract readers.
From my quick literature review, I learnt that extracellular levan is not toxic. The paper cited below mentions that the toxicity is due to the activity of enzyme in cytoplasmic and periplasmic spaces. Majority of the activity of native levansucrase is observed in periplasm. I suggest the author to consider these observations as possibilities that, the enzyme probably became active at lower temperatures and killed the bacteria while in the cyto-/peri-plasm itself (i.e. before translocated into extracellular space).
https://doi.org/10.1128/aem.64.9.3180-3187.1998
: Thanks for the comment. We realize that our use of the term “extracellular space” may have been misleading. In our manuscript, we referred to the periplasmic space as “extracellular,” since levansucrase is translocated across the inner membrane via its signal peptide. However, we agree that the periplasm is distinct from the true extracellular space, and we revised the manuscript to clarify this point and avoid confusion.
Page 7, Line 284: “…ability to undergo reversible transitions between water-soluble and insoluble states”. I recommend the author to change the specific word “insoluble”, since water insoluble states can be aggregates or inclusion bodies as the author mentioned at other places in the manuscript. If the protein was insoluble, changing the temperature from 37 to 16 C would not improve its solubility (insoluble versus ‘solvated, but coacervated or phase-separated’).
: Thanks for the comment. We agree that “insoluble” may misleadingly suggest irreversible aggregation such as inclusion bodies. We have revised the wording to “phase-separated coacervate state” to better reflect the reversible nature of ELP aggregation.
Page 7, Lines 308-310: The observation that the absorbance at 400 nm is higher even when the appearance is not different is interesting. A simple measurement of the UV-vis spectrum of this solution would provide a clearer answer. If a scatter is seen, that would be a direct indication of insoluble materials. I recommend the author to pursue this, and include the data in supporting information.
: Thanks for the comment. We fully agree that UV-Vis spectral analysis can help clarify whether the observed turbidity arises from light scattering. The samples from days 0, 1, 2, 3, 4, and 6 were originally analyzed across the full UV-Vis range (300–700 nm), and the OD400 values reported in the main text were extracted from these spectra. In response to your comment, we revisited the spectral data and extracted the OD600 values from each time point to evaluate wavelength dependence.
As shown in Supplementary Figure S1, the OD600 values exhibit a trend similar to OD400 (Fig. 3a), increasing over time despite no visible turbidity. This wavelength-independent increase strongly supports the presence of light-scattering colloidal structures, such as levan aggregates, rather than true molecular absorption. To provide transparency, we also include the full raw spectral data for all time points as a separate Excel file (Supplementary File 2). We added the following: “To further examine the wavelength dependence of turbidity, we also extracted the OD600 values from the same UV-Vis spectra and plotted them in Supplementary Figure S1. The OD600 trend closely mirrored that of OD400 (Fig. 3a), reinforcing the interpretation that the observed increase in optical density originates from light scattering by colloidal levan aggregates. The full spectral data for all time points are provided in Supplementary File 2 (Excel format).”
Figure 3a: Y-axis label is missing. Also, the author should discuss why the enzyme takes days to show its activity (low kinetics versus lack of molecular motion?). If there is literature precedence for the kinetics of this enzyme (Michaelis-Menten), I recommend that it should be discussed in the manuscript.
: Thanks for the comment. We corrected the figure as commented. Based on our interpretation, the delayed onset of turbidity is not due to intrinsically slow enzymatic kinetics. In fact, published Michaelis-Menten parameters for Bacillus subtilis levansucrase report a KM of ~30mM and a kcat of ~135 s⁻¹, indicating robust catalytic activity under standard conditions (S. Gao, S. Yao, D. J. Hart, Y. An, Biochemical Engineering Journal. 122, 71-74(2017)). We therefore believe that the delay arises from the initially low concentration of active levansucrase in the crude cell lysate. Over time, polymeric levan accumulates gradually until it reaches a threshold level sufficient to scatter light and generate detectable turbidity. This accumulation-driven turbidity is consistent with the time course shown in Fig. 3a and Supplementary Figure S1. We added the following: “The extended lag in turbidity development is unlikely due to limited catalytic efficiency, as B. subtilis levansucrase has been reported to exhibit Km ≈ 30 mM and kcat ≈ 135 s-1.30 Instead, we attribute the delay to the low enzyme concentration in the crude lysate, requiring extended time for polymeric levan to accumulate to colloidal concentrations sufficient for light scattering. This accumulation-dependent threshold effect is consistent with the observed kinetics of optical density increase.”
Figure 3b: It seems that the absorbance at 540 for glucose production is low. A control enzyme (which is either purchased, or made in other hosts) shoud be used to compare the activity. This also tells whether the ELP-induced coacervation retained the structure and function of the enzyme, what percent of the enzyme activity is lost due to this process etc.
: Thanks for the suggestion regarding the use of a control enzyme for activity comparison. We agree that direct benchmarking of the coacervated levansucrase against a non-tagged or commercially available counterpart would help quantify the impact of ELP-induced coacervation on enzymatic function. However, this comparison requires new expression and purification efforts beyond the current scope. We consider this an important direction for future work, where we aim to quantify the retained activity of the ELP-tagged enzyme relative to its native form and systematically assess the structural integrity under phase-separated conditions. We added the following: “Although the absorbance values at 540 nm were relatively low, the difference between SacB-I48 and control samples was consistent and reproducible, indicating specific enzymatic activity. This likely reflects the low concentration of active levansucrase in the crude lysate rather than diminished catalytic efficiency. Importantly, this result supports that the enzyme retains at least partial activity in the coacervated state, suggesting that substrate access and turnover remain feasible even under phase-separated conditions.
While these findings confirm that the ELP-tagged levansucrase is functionally active, a direct comparison of its catalytic performance to a non-tagged or commercially available enzyme would be necessary to quantify any loss of activity due to coacervation. This will be addressed in future studies to better understand how phase separation influences enzymatic function, with planned investigations focusing on catalysis in the condensed or phase-separated state.”
Round 2
Reviewer 1 Report
Comments and Suggestions for Authors
The manuscript has been improved for revision, but some issues remained.
Table 1 and Figure 1 represent the same set of data. Please remove Table 1.
Figure 1: why the E. coli cells grew much better at 16°C than 37°C on the sucrose-free plates?
“B. subtilis”, “Km”, “kcat” should be italic.
Line 392: literature should be cited for “This is consistent with previous reports showing tunable ELP…”
Labels of “a” and “b” are missing in Figure 3.
Author Response
The manuscript has been improved for revision, but some issues remained.
Table 1 and Figure 1 represent the same set of data. Please remove Table 1.
: Thanks for the comment. I have removed Table 1 as advised, and updated the main text accordingly.
Figure 1: why the E. coli cells grew much better at 16°C than 37°C on the sucrose-free plates?
: Thanks for the comment. The cells at 16 °C were grown for 5 days, whereas those at 37 °C were grown overnight. This information has now been explicitly included in the caption for Figure 1.
“B. subtilis”, “Km”, “kcat” should be italic.
: Thanks for the comment. These terms have been italicized as advised.
Line 392: literature should be cited for “This is consistent with previous reports showing tunable ELP…”
: Thanks for the comment. I have now cited the appropriate reference in the text. As commented by Guo et al. (ref #13), this tunability stems from the modular structure of ELPs, with the guest residue (X) and repeat length (n) serving as genetic handles to adjust the transition temperature: more hydrophobic X or longer repeats generally lower the Tₜ, while hydrophilic/charged residues raise it.
Labels of “a” and “b” are missing in Figure 3.
: Thanks for the comment. I have now combined the previous Figures 3a and 3b into a single Figure 3, and labeled the panels as “a” and “b” accordingly.
Reviewer 2 Report
Comments and Suggestions for Authors
I thank the author for considering my suggestions, and for making appropriate changes. I still believe that better conclusions can be made with a control soluble protein, I highly recommend including it in the subsequent studies. Both the questions regarding the ability of enzyme to retain its native fold, and the slow rate of turbidity can be answered easily with the help of this control. However, I am now accepting the paper for publication.
Author Response
I thank the author for considering my suggestions, and for making appropriate changes. I still believe that better conclusions can be made with a control soluble protein, I highly recommend including it in the subsequent studies. Both the questions regarding the ability of enzyme to retain its native fold, and the slow rate of turbidity can be answered easily with the help of this control. However, I am now accepting the paper for publication.I believe that this study and the technique developed will have an impact on the field. However, I also believe (for the following reasons) that the study is incomplete and can answer more questions using simple experiments. Additionally, the manuscript may be rephrased to reduce the word-count as longer texts tend to distract readers.
: Thanks for the comment. I sincerely appreciate the reviewer’s recognition that this work has the potential to make an impact in the field. I also acknowledge the reviewer’s valuable suggestion regarding the inclusion of a control soluble protein, which would indeed strengthen the conclusions. I humbly accept the point that this study is part of an ongoing effort and should be further developed in future work.